# Treatment and Prevention of Histoplasmosis in Adults Living with HIV

**DOI:** 10.3390/jof7060429

**Published:** 2021-05-28

**Authors:** David S. McKinsey

**Affiliations:** Metro Infectious Disease Consultants, Kansas City, MO 64132, USA; David.mckinsey@hcamidwest.com; Tel.: +1-816-444-7977; Fax: +1-816-361-8938

**Keywords:** histoplasmosis, HIV, amphotericin, itraconazole, fluconazole, voriconazole, posaconazole, prophylaxis, opportunistic infections, subaitraconazole

## Abstract

Histoplasmosis causes life-threatening disseminated infection in adult patients living with untreated HIV. Although disease incidence has declined dramatically in countries with access to antiretroviral therapy, histoplasmosis remains prevalent in many resource-limited regions. A high index of suspicion for histoplasmosis should be maintained in the setting of a febrile multisystem illness in severely immunosuppressed patients, particularly in persons with hemophagocytic lymphohistiocytosis. Preferred treatment regimens for initial therapy include liposomal amphotericin B for severe disease, or itraconazole for mild to moderate disease. Subsequently, itraconazole maintenance therapy should be administered for at least one year and then discontinued if CD4 count increases to ≥150 cells/µL. Antiretroviral therapy, which improves outcome when administered together with an antifungal agent, should be instituted immediately, as the risk of triggering Immune Reconstitution Syndrome is low. The major risk factor for relapsed infection is nonadherence. Itraconazole prophylaxis reduces risk for histoplasmosis in patients with CD4 counts <100/µL but is not associated with survival benefit and is primarily reserved for use in outbreaks. Although most patients with histoplasmosis have not had recognized high-risk exposures, avoidance of contact with bird or bat guano or inhalation of aerosolized soil in endemic regions may reduce risk. Adherence to effective antiretroviral therapy is the most important strategy for reducing the incidence of life-threatening histoplasmosis.

## 1. Introduction

*Histoplasma capsulatum*, the most common endemic fungus in the United States [1], causes life-threatening disseminated disease in immunosuppressed patients living with HIV infection [2,3,4]. Although the availability of highly active antiretroviral therapy (ART) led to a dramatic reduction in the incidence of histoplasmosis in the HIV population in the United States [5], in some areas in Central and South America, disease remains widely prevalent and is a leading cause of mortality [6]. Histoplasmosis also remains a threat to persons living with HIV in developed countries who either lack access to ART or are nonadherent [7].

This review will address the pathophysiology, clinical manifestations, and prevalence of histoplasmosis in adults living with HIV; describe the experience with various treatment regimens and discuss the antifungal regimens currently recommended by guidelines from public health agencies and professional societies; discuss the management of histoplasmosis in the setting of immune reconstitution; and review the recognized risk factors for histoplasmosis and strategies for disease prevention.

## 2. Pathophysiology

Inhalation of airborne *H. capsulatum* microconidia, after soil disruption or other activities that disperse organisms into the atmosphere, is the first stage of infection [8].

After being engulfed by phagocytic cells, microconidia warm to body temperature and undergo intracellular conversion to yeast, the pathologic form of the organism [9]. Yeast multiply within airspaces and then spread to adjacent alveoli and subsequently to hilar and mediastinal lymph nodes. In the setting of intact cellular immunity, the proinflammatory cytokines released by CD4 Th1 cells, interferon gamma, TNF-alpha, and GM-CSF activate macrophages to kill phagocytosed yeast [9,10].

In the general population, primary *H. capsulatum* infection is asymptomatic in 99% of cases. Acute pulmonary histoplasmosis is the usual manifestation of symptomatic illness. Hematogenous dissemination, although common, is usually self-limiting [8]. However, in the setting of severe CD4 cell deficiency in patients with advanced HIV infection, *H. capsulatum* usually causes life-threatening progressive disseminated disease. In HIV-infected patients, symptomatic illness has been observed in 55% of cases of primary histoplasmosis [11]. Progressive disseminated histoplasmosis (PDH) occurred in 97% to 100% of cases of symptomatic histoplasmosis prior to the availability of highly active ART [3,11] and was associated with reduced life expectancy [11].

The issue of whether disseminated histoplasmosis in immunosuppressed patients occurs due to primary infection, reinfection, or dissemination of latent foci persisting after remote infection remains unsettled and controversial. This issue has broad clinical implications [7,11,12]: if reactivation occurs, a large population of immunosuppressed persons would be at risk; for example, approximately 20% of the U.S. population has had prior subclinical histoplasmosis [13].

Increases in disease incidence during outbreaks have supported the concept that primary infection or reinfection occur [14]. Cases among persons who moved from high-prevalence regions to presumed non-endemic areas have suggested that reactivation is another possible pathophysiologic mechanism [15], a concept supported by the development of a murine model of early reactivation [16]. A prospective study compared disease incidence in patients who had evidence of prior histoplasmosis, defined by reactive Histoplasmin skin tests, pulmonary calcifications, or positive Histoplasma serology, to those without objective evidence of previous infection. New symptomatic or subclinical cases of histoplasmosis occurred in 9.9% of patients in the group with evidence of prior disease and in 4.0% of those without signs of prior infection. Whether cases in the previously infected group were due to secondary infection or reactivation of latent infection could not be ascertained. If reactivation did occur, it was uncommon, as the majority of patients who had evidence of prior histoplasmosis and were severely immunocompromised did not develop symptomatic infection [11]. The infrequency, or perhaps the lack of occurrence, of reactivation in persons living with HIV has been mirrored in the solid-organ transplant population [17]. One possible explanation is that reactivation could occur relatively early after resolution of primary infection, as demonstrated in the mouse model [16], but not at a remote stage after *H. capsulatum* organisms contained within thick-walled calcified granulomas are no longer viable.

## 3. Clinical Manifestations

Two primary forms of PDH have been recognized in the setting of HIV infection: a subacute wasting syndrome associated with fever, chills, and profound weakness, often accompanied by pulmonary infiltrates and diarrhea, and a fulminant presentation associated with multiple-organ system failure [2,3]. A study performed prior to the availability of highly active ART found that the mortality of patients with subacute manifestations was 2%, whereas that of persons with severe disease was 70% [18]. Multivariate analyses in two studies have demonstrated that factors independently associated with severe or fatal disease included renal failure and hypoproteinemia. Receipt of ART decreased risk for severe disease [19,20]. Hemophagocytic lymphohistiocytosis (HLH), manifested by cytokine storm and excessive macrophage activation, likely accounts for many of the fulminant cases of disseminated histoplasmosis in patients with AIDS [21,22]. In French Guiana, 18% of HIV-infected persons hospitalized with histoplasmosis had HLH [23]. HIV is the most common underlying condition in patients with histoplasmosis who have HLH [21,22,23]. Dexamethasone, cyclosporine A, and etoposide have been used for treatment of HLH in other patient populations [24], but the use of these immunosuppressive therapies in persons with advanced HIV is problematic and the risk–benefit ratio is likely unfavorable [21,22].

## 4. Prevalence

Histoplasmosis was one of the most common opportunistic infections in the midwestern U.S. in the 1980s and 1990s, affecting up to 20% of the HIV population in some midwestern cities [25]. In Kansas City, between 1990 and 1993, the annual incidence of symptomatic histoplasmosis was 3% [11]. Histoplasmosis remained prevalent in the early years of the 21st century: among patients with PDH at a U.S. urban teaching hospital from 2000 to 2010, 41% were HIV seropositive [26].

In recent years, the incidence of disseminated histoplasmosis in HIV-infected patients has plummeted in developed countries with widespread access to ART [27]. Elevated HIV viral load is an independent risk factor for opportunistic infections including histoplasmosis [28,29]. In a large prospective cohort study conducted at multiple centers in the U.S. the incidence of histoplasmosis dropped to zero after 2003 [5]. Thus, it seems clear that immune reconstitution and viral load suppression greatly reduce the risk for histoplasmosis. In the period 2018–2019, enhanced surveillance for histoplasmosis in 8 U.S. states showed that less than 3% of identified cases had underlying HIV/AIDS; solid-organ transplantation and autoimmune diseases were much more commonly identified risk factors [30]. However, given that only half of the U.S. population living with HIV are in care and have an undetectable viral load [31] the at-risk population remains high. Cases of histoplasmosis continue to be diagnosed in persons initially presenting with advanced, previously undiagnosed disease [32]. Further, histoplasmosis remains the most common AIDS-defining infection and cause of HIV-related death in French Guiana [33], and more than 40% of febrile patients with HIV in the central and northeast regions of Brazil have histoplasmosis [34].

## 5. Treatment

In the first few years of the AIDS pandemic, histoplasmosis carried high risk for mortality. The use of ketoconazole was abandoned because of lack of efficacy. Amphotericin B was effective initially in many cases but 60% subsequently experienced relapsed infection after treatment was stopped [2]. Long-term maintenance therapy reduced the frequency of relapse [35,36] Thus, two treatment phases are recommended: initial induction and subsequent long-term maintenance therapy.

Only a few prospective clinical trials of induction or maintenance treatment have been undertaken. In patients treated with itraconazole, 200 mg bid for 12 weeks and then 200 mg daily for at least one year, 85% responded [37]. Conversely, in a fluconazole single-arm study the efficacy was substantially lower. In the initial phase of this study, 10 of 20 subjects relapsed; accordingly, the protocol was intensified, with an increase in daily dose from 600 to 800 mg. Subsequently, 74% had a successful outcome [38]. In the only prospective double-blind randomized clinical trial of histoplasmosis treatment ever conducted, 82% of patients responded to liposomal amphotericin B, whereas only 56% who received amphotericin B deoxycholate improved. Further, nephrotoxicity was significantly higher in patients who received the deoxycholate formulation [39]. Antiretroviral therapy improves response rate to antifungal treatment in patients with AIDS and histoplasmosis [40].

Although a recent Cochrane analysis concluded that the optimum maintenance regimen for histoplasmosis has not been determined, as no published study has compared <12 months to >12 months of maintenance treatment [41], several regimens have been effective. In patients treated with itraconazole, 200 or 400 mg daily, following induction therapy, 95% were relapse-free after 1 year [36]. Weekly or biweekly amphotericin B deoxycholate was also effective albeit this treatment has rarely been used because of potential toxicity and the need for intravascular access [35]. Currently, the main predictor of relapse is not the choice of maintenance antifungal regimen, but rather nonadherence with HIV care [42] which remains a widely prevalent and vexing problem [43].

HIV-infected patients with confirmed or suspected histoplasmosis should be presumed to have disseminated disease. Liposomal amphotericin B is the first-line treatment for acutely ill patients until their clinical status improves, usually a week or two. Subsequently, oral itraconazole should be prescribed. Itraconazole is the treatment of choice for mild–moderate disease and for maintenance therapy.

Absorption of itraconazole is pH dependent and varies widely from patient to patient [44]. Drug level monitoring is recommended, as serum concentrations can be subtherapeutic due to absence of gastric acidity, nonadherence, or drug–drug interactions. Authorities recommend that random itraconazole levels, after steady state equilibration, should be maintained in the 1–2 µg/mL range, a level far exceeding the *H. capsulatum* MIC_90_ of 0.06 µg/mL. The sum of itraconazole and hydroxyitraconazole, its active metabolite, should be used for this calculation [7,45,46].

In light of the variability of absorption of itraconazole capsules, a new formulation, subaitraconazole, has been developed. Subaitraconazole contains the parent compound dispersed in a polymeric matrix targeted to improve absorption in the upper small intestine. Recent crossover clinical trials compared subaitraconazole to itraconazole and determined that the bioavailability of subaitraconazole was 173% higher, and patient to patient variability of serum concentrations was substantially lower, than that of itraconazole [44]. In one study subaitraconazole was more likely to attain the target serum concentration [≥1 µg/mL] than itraconazole (81% vs. 44%) [47]. A subaitraconazole dose of 65 mg is equivalent to a 100 mg itraconazole dose. A prospective randomized clinical trial comparing itraconazole to subaitraconazole for histoplasmosis and other endemic mycoses was completed by the Mycoses Study Group in 2021; data have not yet been released as of the publication date of this manuscript. If subaitraconazole is not available, liquid itraconazole is preferred if tolerated because of improved properties of absorption, presuming that cost is not prohibitive.

In a prospective study assessing discontinuation of prophylaxis after successful immune reconstitution from antiretroviral therapy, defined as a sustained CD4 count >150/µL, no relapses occurred among 32 patients studied [48]. Criteria for discontinuation of secondary prophylaxis include receipt of antifungal therapy for at least 12 months and antiretroviral therapy for at least 6 months. Prophylaxis should not be discontinued in nonadherent patients or those with central nervous system histoplasmosis [7].

## 6. Alternative Regimens

Fluconazole, which has lower in vitro activity against *H. capsulatum* than itraconazole, has been less effective in clinical trials and animal models. Fluconazole use was associated with a slower decline in antigenuria in humans than itraconazole [49]. Resistance to fluconazole emerged during treatment of disseminated histoplasmosis in multiple patients with HIV/AIDS [50]. Among patients who did not respond to fluconazole 71% of post-treatment isolates had 4-fold or higher increases in minimum inhibitory concentrations (MICs) [50]. Resistance was linked to a single point mutation (Y136F) in the cp450 alpha demethylase [50]. Given fluconazole’s lower efficacy than itraconazole and the high risk for development of resistance, its use for induction or maintenance therapy is not recommended [41].

Limited experience with the potential alternatives voriconazole, posaconazole, and isuvaconazole has been reported. The median MICs for these 3 drugs for *H. capsulatum* isolates collected in a clinical trial were 0.015 µg/mL, ≤0.007 µg/mL, and ≤0.007 µg/mL, respectively, compared to 1.0 µg/mL for fluconazole [51]. Fluconazole resistance was associated with cross resistance to voriconazole in 41% of cases [51], but not to isuvaconazole or posaconazole [51,52]. A single center retrospective review of 261 patient with histoplasmosis, one-fourth of whom had HIV/AIDS, assessed outcomes after initial azole treatment with either itraconazole or voriconazole. In this study, the use of voriconazole was associated with significantly higher mortality in the first 6 weeks [53].

Posaconazole, which has high in vitro activity against *H. capsulatum,* was more effective than itraconazole in an animal model of histoplasmosis [54] and has been effective for salvage therapy in a few cases [55]. No prospective trials of posaconazole therapy for histoplasmosis have been published. The role of therapeutic drug monitoring has not been defined fully but experts have suggested aiming for a trough serum posaconazole concentration above 0.5 µg/mL [46]. Isuvaconazole appears to have a higher barrier to resistance than voriconazole and exhibits less pharmacokinetic variability. Published experience with the use of isuvaconazole for HIV-infected patients with histoplasmosis has been extremely limited [56].

The triazole antifungal drugs inhibit certain cytochrome p450 isoenzymes to varying degrees and thus increase the effects of other drugs metabolized by these enzymes. Further, the concomitant use of medications that induce cytochrome p450 isoenzymes causes accelerated clearance of triazoles and can lead to subtherapeutic serum concentrations of antifungal medications, potentially contributing to treatment failure. As an example, rifampin substantially increases clearance of all triazole antifungal drugs. In the setting of HIV infection multiple important interactions between antiviral and antifungal medications can occur. Serum concentrations of protease inhibitors are increased by itraconazole, fluconazole, posaconazole, and voriconazole [57]; conversely, isuvaconazole can reduce levels of lopinavir/ritonavir [58]. Efavirenz coadministration reduces serum concentrations of itraconazole, voriconazole, and posaconazole. The NIH Clinical Guidelines provide a detailed description of clinically significant antifungal drug interactions [57].

The echinocandins do not have in vitro activity against the yeast form of *H. capsulatum*; caspofungin and micafungin were ineffective in animal models [59,60] Thus, echinocandins are not recommended for treatment of histoplasmosis. Several other widely available medications which have in vitro activity against *H. capsulatum* but have not been studied in animal models or humans include trimethoprim-sulfamethoxazole, ciprofloxacin, isoniazid derivatives [61] and ritonavir, which is 7-fold more effective against the yeast form than mycelia [62].

## 7. IRIS

Immune reconstitution inflammatory disorder (IRIS) can occur following initiation or resumption of antiretroviral therapy in severely immunocompromised HIV-infected patients with symptomatic or subclinical histoplasmosis. IRIS is manifested either by a paradoxical flare of known infection or by unmasking of previously unrecognized disease and is most often associated with prolonged fever [63]. Features consistent with the diagnosis of histoplasmosis IRIS include apparent worsening disease in the face of therapeutic serum concentrations of itraconazole and declining antigenemia or antigenuria; culture negativity despite visualization of yeast in tissue specimens; and response to immunosuppressive therapy [64]. A retrospective study in French Guiana identified the rate of histoplasmosis IRIS to be 0.74/1000 person-years [63]. The histoplasmosis IRIS case rate was substantially lower than had reported with *M. tuberculosis* and cryptococcal meningitis [65]. The median time to onset of IRIS was 11 days [63].

The optimum management of IRIS has not been defined. As IRIS is self-limiting, experts recommend that antiretroviral and antifungal treatment should be continued. The role of steroid therapy, which could potentially exacerbate histoplasmosis, has not been established [63]. The available histoplasmosis treatment guidelines, discussed below, do not make recommendations to treat IRIS with anti-inflammatory medications. In cases of life-threatening IRIS it is reasonable to begin a corticosteroid if the patient is receiving effective antifungal therapy; brief treatment duration at the lowest possible dose seems advisable. A randomized clinical trial which compared early (within the first 2 weeks after diagnosis of HIV infection) versus deferred antiretroviral therapy (ART) found that early ART was associated with less AIDS progression and death; in this study, in which a limited number of cases of histoplasmosis were identified, timing of initiation of ART was not linked to histoplasmosis IRIS [66]. Thus, antiretroviral therapy does not need to be delayed in newly diagnosed cases of advanced HIV infection with histoplasmosis.

## 8. Treatment Guidelines

Treatment guidelines for histoplasmosis have been published by multiple organizations: the Infectious Diseases Society of America (IDSA), the National Institutes of Health (NIH) and the Centers for Disease Control and Prevention (CDC) in collaboration with the HIV Medical Association (HIVMA) and the IDSA; the American Thoracic Society; and the World Health Organization (WHO)/Pan American Health Organization (PAHO) [46,67,68,69]. Recommendations from the four sets of guidelines are similar. A one-to-two week course of induction therapy is recommended by the IDSA, with liposomal amphotericin B, 3 mg/kg/day for severe disease or itraconazole, 200 mg bid for 12 months following a 3-day loading dose of 200 mg tid, for less acutely ill patients. (Table 1) Long-term maintenance itraconazole therapy, 200 mg daily, should then be administered for a minimum period of 12 months. Levels of antigenuria and antigenemia should be monitored for the first year; increasing antigen values raise the possibility of treatment failure or impending relapse. Other lipid amphotericin formulations or amphotericin B deoxycholate are potential alternative options [46,67]. The CDC/NIH and WHO/PAHO guidelines recommend induction for 2 weeks rather than the 1 to 2 week period advised by the IDSA [66,68]. Alternative azole options recommended by the NIH/CDC include posaconazole 300 mg bid for 1 day, then 300 mg daily; voriconazole 400 mg bid for 1 day, then 200 mg bid; or the least desirable option, fluconazole 800 mg daily [67]. For long-term suppressive therapy, each of the four guidelines recommend itraconazole. A dose 200 mg daily is advised by IDSA whereas the CDC/NIH recommend 200 mg bid. Alternative options in rank order include posaconazole 300 mg extended release capsule daily; voriconazole 200 mg bid; or fluconazole 400 mg daily (Table 2) [46,67,68,69]. After successful immune reconstitution, defined as CD4 count increase to 150/µL or higher, secondary prophylaxis can be stopped once the patient has received a minimum 6 month course of ART and at least 12 months of antifungal therapy, has serum and urine antigen levels <2, and has negative blood cultures [46,67]. Antigen may be detected at low levels for prolonged periods of time despite clinical improvement.

Recent studies provided insight into antifungal prescribing practices in the U.S. Among thousands of cases of histoplasmosis identified in the MarketScan database, most of whom were outpatients, 40% who received oral therapy were treated with either fluconazole, voriconazole, or posaconazole. A second study which detected hundreds of cases by enhanced surveillance determined that 15% received one of these three alternative regimens. The numbers of patients who were HIV seropositive in these two studies were not specified. The relatively common practice of using non-preferred regimens suggests that clinicians may not be routinely following IDSA treatment guidelines [70,71].

## 9. Risk for Histoplasmosis

Strategies for prevention of histoplasmosis in HIV-seropositive patients have been devised, based on identification of risk factors, assessment of the efficacy of itraconazole prophylaxis, analysis of outbreaks in the general population, and redefinition of the geography of the endemic area.

In the early years of the AIDS pandemic, most diagnosed cases of histoplasmosis were identified in the traditional endemic region in the U.S. A prospective study of HIV-infected patients in Missouri showed that CD4 lymphocyte counts <300/μL significantly increased risk for histoplasmosis, and that risk increased sharply in the subgroup with CD counts <150/μL [11]. Other independent risk factors included chicken coop exposure and positive baseline complement fixation mycelial serology. Rural residence, positive Histoplasmin skin test, positive complement fixation yeast serology, and pulmonary calcifications on chest imaging studies were not associated with higher incidence of disease [11]. A subsequent multicenter case control study showed that the risk for histoplasmosis was higher in individuals who had handled soil contaminated with bird or bat guano and was significantly lower in those who had received antiretroviral therapy or had been treated with an azole drug within the past 2 months. Use of trimethoprim-sulfamethoxazole for Pneumocystis prophylaxis also reduced the risk for histoplasmosis [72]. A large retrospective cohort study in French Guiana found that CD4 count <200/µL; CD4 nadir <50/µL; and receipt of antiretroviral therapy for <6 months were associated with higher risk. Factors independently associated with decreased risk included prior treatment with fluconazole; antiretroviral therapy for >6 months; or *Pneumocystis jiroveci* pneumonia [73]. The best objective marker for predicting high-risk patients is CD4 count <150/µL; elevated HIV viral load is also almost certainly associated with higher risk.

## 10. Prophylaxis

Fluconazole prophylaxis did not reduce risk for histoplasmosis in a placebo-controlled randomized multicenter study in which most sites were not located in the endemic region [74]. Itraconazole prophylaxis was studied in a prospective randomized double-blind placebo-controlled study in 4 U.S. cities in the endemic area. In the subset of patients with CD4 counts <100/µL, itraconazole significantly reduced cases of histoplasmosis and cryptococcosis but did not improve survival [75]. Itraconazole prophylaxis, which appeared to induce resistance among colonizing Candida species [76], was well tolerated. However, triazole prophylaxis does carry a risk for toxicities, drug–drug interactions, and QT interval prolongation. In light of the decline in incidence of histoplasmosis in patients receiving ART in developed counties, itraconazole prophylaxis is not used routinely. IDSA and NIH/CDC guidelines recommend consideration of prophylaxis during a histoplasmosis outbreak or in settings in which the disease incidence is >10 cases per 100 patient-years for persons with CD4 <150/µL [46,67].

Given that receipt of trimethoprim sulfamethoxazole (TMP/SMX), which has in vitro activity against *H. capsulatum,* was associated with lower disease incidence in the case control study [72], and that patients with Pneumocystis infection, which presumably was treated with TMP/SMX, in French Guiana had a lower incidence of histoplasmosis [73], it is plausible that TMP/SMX could reduce risk for histoplasmosis; further investigation will be necessary to assess this hypothesis.

## 11. Non-Pharmacologic Strategies for Prevention

Assessment of histoplasmosis outbreaks and enhanced surveillance in the general population have informed the approach to disease prevention. Findings from these investigations may be extrapolated to provide recommendations to reduce histoplasmosis risk for persons living with HIV. Outbreaks have been linked to cave or tunnel exploration; cleaning or clearing bird roosts, old chicken coops, parks, or school courtyards; demolition; excavation; exposure to chicken coops or decayed trees; use of bird or bat guano for fertilizer; and sawing rotten wood [26,77]. A survey of 105 outbreaks from 1983 to 2013 identified that exposure to birds, bats, or their droppings was reported in more than three-fourths of cases [78].

Certain vocations and hobbies increase the risk of histoplasmosis. The relative risk for agricultural or forestry workers, fishermen, or hunters developing histoplasmosis is almost 10-fold higher than that of the general population, and persons engaged in construction or extraction activities have almost a 5-fold increased relative risk. In general, activities associated with soil aerosolization and bird or bat guano exposure have been the most commonly identified causes of outbreaks and occupationally acquired infections [30].

The U.S. Centers for Disease Control recommends that the following measures be undertaken to reduce the risk for histoplasmosis in HIV-infected patients with CD4 cell counts <150 cells/μL who reside in the endemic area: avoid disturbing soil contaminated with chicken or bird guano; spelunking; remodeling, demolishing or cleaning old buildings; or cleaning chicken coops [67]. Other precautions advised by the National Institute for Occupational Safety and Health for the general public, in the setting of recognized high-risk environmental exposure, include use of a respirator and wetting potentially contaminated materials, then placing in an impervious bag and disposing in a landfill [79].

It is worth noting that the traditional histoplasmosis endemic area in the U.S. has been expanding north and westward, thus increasing the size of the at-risk population. Further, the worldwide distribution of authochthonous cases is much more widespread than previously recognized; in recent years, cases have been identified from regions as diverse as the Yangtze River region of China; Thailand; the southern half of Africa; and several other areas on all continents with the sole exception of Antarctica [80]. Thus, it seems advisable for persons living with HIV to exhibit caution when engaging in known high-risk activities for histoplasmosis exposure even if they do not reside in a traditional endemic region.

## 12. Conclusions

Dramatic progress has been made in the treatment and prevention of histoplasmosis in adult patients living with HIV. The foremost need for immunosuppressed HIV-infected persons is for effective ART, which both reduces the risk for histoplasmosis and improves treatment outcome. Among severely immunocompromised individuals who delay seeking medical attention, are nonadherent, or reside in resource-limited regions with lack of ready access to ART, a high index of suspicion for histoplasmosis should be maintained in the setting of a febrile multisystem illness, and a preferred treatment regimen, with liposomal amphotericin B or itraconazole, should be initiated in accordance with evidence-based guidelines. Mitigation of high-risk exposures and/or use of itraconazole prophylaxis in certain patient subsets are strategies likely to reduce the incidence of life-threatening histoplasmosis.

## Figures and Tables

**Table 1 jof-07-00429-t001:** Induction therapy for histoplasmosis in adults with HIV *.

Drug	Indication	Dose	Duration	Resistance	Evidence
Itraconazole	Mild–moderate	200 mg tid for 3 d, then 200 mg bid	12 months	No	Prospective single-arm study
Liposomal amphotericin	Moderate or severe	3–5 mg/kg/d	1–2 weeks	No	Prospective randomized double-blind study
Fluconazole	Not recommended			Yes	Prospective single-arm study

* Antiretroviral therapy is also recommended.

**Table 2 jof-07-00429-t002:** Maintenance therapy for histoplasmosis in adults with HIV *.

Drug	Dose	Duration	Resistance	Evidence	Comment
Itraconazole	200 mg bid	≥12 months	No	Prospective single-arm study	Drug of choice Liquid formulation preferred if available
Posaconazole	300 mg extended release daily	≥12 months	No	in vitro and animal model data; case reports	Limited experience
Voriconazole	200 mg bid	≥12 months	Yes	in vitro data; single small case series	Less effective than itraconazole
Fluconazole	400 mg daily	≥12 months	Yes	Prospective single-arm study	High risk of failure due to emergence of resistance
Amphotericin deoxycholate	50 mg q 2 weeks	≥12 months	No	Single arm trial	IV required; potential for toxicity

* Antiretroviral therapy is also recommended.

## Data Availability

Not applicable.

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
