# Peer review of "Treatment and Prevention of Histoplasmosis in Adults Living with HIV"

_jof, 2021, doi:10.3390/jof7060429_

Round 1

Reviewer 1 Report

This is a very nice review of the incidence and management of histoplasmosis in HIV patients. This is an entity that is being observed less and less in the era of widely available ART, however, as correctly identified by the author, histoplasmosis remains an issue in resource limited settings. Comments on the manuscript are outlined below:

1) Line 18: It should be mentioned here that itraconazole prophylaxis is not widely recommended and indicated in only very specific situations (such as an outbreak).

2) Line 20: It should be stated that most patients diagnosed with histoplasmosis have no specific exposure. It would not be prudent to mislead clinicians into not considering histo in situations where a specific exposure is not reported.

3) Line 92: wording is confusing here - should eliminate semicolon and start a new sentence to state: receipt of ART and reconstitution of immune function decreases risk

4) Line 94: It would be helpful to mention exactly how many cases of histo in HIV patients present with HLH. This is not something that seems to be all that common.

5) Line 98: should specifically state that in the setting of disseminated histo in an HIV infected patient with AIDS, that immune suppressive therapies may be problematic since these treatments certainly may be of clinical benefit in other cases of HLH

6) Line 111: should state that the combination of viral suppression plus immune reconstitution will reduce risk of histo

7) Line 114: The authors individual personal experience, although interesting, is not relevant to such a review where a premium should be placed on population level data.

8) Line 173: It should also be stated that if subitraconazole is not available, liquid itraconazole is preferred if tolerated because of improved properties of absorption

9) Line 223: should state...Apparent worsening of disease...since the symptoms are related to inflammation and not actual worsening of histo infection.

10) Line 232: Should provide some guidance here on when to consider use of a corticosteroid, for example, in the setting of life threatening IRIS symptoms.

11) Line 265: Should mention that antigen levels may not become negative for a very prolonged period of time.

12) Line 365, Table 2: for the comment section for itraconazole should mention that the liquid or subitraconazole formulation is preferred over capsules because of better absorption.

13) Line 368, Table 3: Could be eliminated since this information is nicely summarized in the text.

14) Major criticism: There should be a section outlining drug interactions of itraconazole, posaconazole, isavuconazole with HIV medications. There are several interactions that occur because of cytochrome P450 metabolism issues especially with protease inhibitors.

Author Response

The reviewer’s thoughtful comments are greatly appreciated. The manuscript has been revised to incorporate these suggestions.

  1. The abstract has been revised to make the point that itraconazole prophylaxis is recommended only in certain rare situations.
  2. The abstract has been revised to make the point that most cases of histoplasmosis are not linked to specific exposures.
  3. Sentence rewritten as suggested
  4. The facts that 18% of hospitalized patients with histoplasmosis have HLH, and that HIV is the most common underlying disorder in patients with histoplasmosis who have HLH, have been added to the discussion.
  5. Sentence reworded to make the point that immune suppressive therapies are problematic in the setting of HIV infection.
  6. Sentence reworded to make the point that the combination of viral suppression plus immune reconstitution reduces the risk of histoplasmosis
  7. Sentence omitted
  8. Sentence added to make the point that liquid itraconazole is preferred if available and if not cost-prohibitive (note the price in the US is 9-fold higher than the capsule formulation – however, I did not mention specific costs in the manuscript.)
  9. The word apparent has been added
  10. Sentence revised to make the point that steroid therapy is reasonable for life-threatening IRIS but that none of the histoplasmosis guidelines make specific treatment recommendations for IRIS.
  11. Sentence revised to make the point that antigen may remain detectable for prolonged periods
  12. Done; see #8
  13. Tables 3 and 4 have been eliminated
  14. New paragraph added discussing drug-drug interactions with the triazoles

Reviewer 2 Report

This is an excellent review of treatment and prevention of histoplasmosis. I have only a few little things to point out.

Abstract: line 11 "illness" not illnesses"

clinical manifestations: line 90 "multivariate" or "multivariable", not "multivariant"

Alternative regimens: line 184  "...associated with a slower..."

Non-pharmacologic strategies for prevention: line 340 "bird", not "bid"

In that same section, perhaps you should mention mediation suggested by NIOSH for those involved in demolition work. I have added the reference. This is listed in Table 4, but not in the text. (Lenhart SW, Schafer MP, Singal M, et al: Histoplasmosis: Protecting workers at risk.  DHHS (NIOSH) Publication No. 97-146, 1997)

Tables 3 & 4: These are very hard to read and look rather messy.  Perhaps the journal could help make them more readable. Also, not sure why several items in Table 4 have asterisks, and also why low CD4s are listed under behavioral subheading

Author Response

The reviewer’s thoughtful comments are greatly appreciated. The manuscript has been revised to incorporate these suggestions.

  1. Line 1 edit made as suggested
  2. Line 90 correction made
  3. Line 184 edit made as suggested
  4. Line 340 correction made
  5. Sentences added to address mitigation strategies recommended by NIOSH; citation added
  6. Tables 3 and 4 have been omitted at the suggestion of the other reviewer. Note the “messiness” resulted from a change in font by the publisher.

Round 2

Reviewer 1 Report

The revised manuscript addresses previous criticisms well and is comprehensive. I have no other critical comments and enjoyed reviewing this manuscript.